# Prediction of Bone Healing around Dental Implants in Various Boundary Conditions by Deep Learning Network

**DOI:** 10.3390/ijms24031948

**Published:** 2023-01-18

**Authors:** Pei-Ching Kung, Chia-Wei Hsu, An-Cheng Yang, Nan-Yow Chen, Nien-Ti Tsou

**Affiliations:** 1Department of Materials Science and Engineering, National Yang Ming Chiao Tung University, Hsinchu 30010, Taiwan; 2National Center for High-Performance Computing, Hsinchu 30076, Taiwan

**Keywords:** deep learning, data-driven, tissue differentiation, bone healing

## Abstract

Tissue differentiation varies based on patients’ conditions, such as occlusal force and bone properties. Thus, the design of the implants needs to take these conditions into account to improve osseointegration. However, the efficiency of the design procedure is typically not satisfactory and needs to be significantly improved. Thus, a deep learning network (DLN) is proposed in this study. A data-driven DLN consisting of U-net, ANN, and random forest models was implemented. It serves as a surrogate for finite element analysis and the mechano-regulation algorithm. The datasets include the history of tissue differentiation throughout 35 days with various levels of occlusal force and bone properties. The accuracy of day-by-day tissue differentiation prediction in the testing dataset was 82%, and the AUC value of the five tissue phenotypes (fibrous tissue, cartilage, immature bone, mature bone, and resorption) was above 0.86, showing a high prediction accuracy. The proposed DLN model showed the robustness for surrogating the complex, time-dependent calculations. The results can serve as a design guideline for dental implants.

## 1. Introduction

The bone healing process is difficult to be observed in clinical experiments. In the past, the healing performance of dental implants could only be observed through histomorphologic images [1] and CT images [2]. However, the methods cannot continuously observe the process of bone growth due to the high cost, as well as they often require the sacrifice of animals. Thus, finite element (FE) analysis and artificial intelligence/deep learning network (AI/DLN) have been used to examine dental implants and bone healing, achieving the purpose of prediction before insertion surgery so as to reduce the cost of trial and error [3]. For example, DLNs were applied to classify dental implant systems by data of periapical radiographs [4,5]. Huang et al. [6] developed a DLN to predict the 5-year implant loss risk based on cone-beam computed tomography images. Celik [7] used panoramic radiographs as datasets to enable AI to detect dental conditions, and thus, the corresponding treatment can be well planned. On the other hand, numerical empirical equations were proposed to describe the behavior of bone ingrowth [8,9] as an alternative. The mechano-regulation algorithm is one of the commonly used approaches to predict tissue differentiation in the short-term bone healing process [10,11], which reveals the effect of implant geometry on the performance of bone healing, giving the design guidelines for the implant’s geometry [1]. Various implant designs are used in the literature, including features such as platform-switched, double-threaded, and tapered body. The design of each implant has its benefits and drawbacks. For example, deeper threads ensure increases in the surface for attaching the bones; tapered implants allow for easy insertion and create excellent primary stability by decreasing the stress on the surrounding bone and by minimizing the crestal bone loss [12,13]. It is also observed in clinical, animal, and finite element analysis studies that the initial bone density and occlusal force have a great influence on bone healing indexes such as implant stability quotient (ISQ) [14], peri-implant bone loss [15], and bone remodeling [16]. This indicates that the boundary conditions in the tissue differentiation prediction methods, such as mechano-regulation theory, need to be taken into account.

However, to the best of our knowledge, a systematic analysis for the bone healing process subjected to various implants and health conditions of patients, e.g., age, gender, occlusal force, is still absent. Although numerical methods allow for observing step-by-step dynamic changes revealing the underlying features, the major limitation is the demand for rising computing resources, where the calculation of the tissue differentiation process usually takes hours on a standard desktop computer. A typical solution for the above-mentioned problem is to build a neural network (NN) model to substitute and accelerate the cumbersome calculations. Several studies have been dedicated to developing NN models as surrogates for finite element analysis (FEA) [17,18,19]. However, most of the NN-based models were designed for predicting the mechanical responses, such as stress, strain, and temperature, and focus on the specific boundary conditions. For problems involving complex physical mechanisms, case-dependent boundary conditions, and material properties, it is not straightforward for a single NN model to replace the entire FEA since the wide variation in the boundary condition cannot be fully covered by the training dataset. Thus, multiple DNN structures may be needed to decompose the complex calculation procedures; a well-designed input format should also be sufficiently general and representative of the boundary conditions.

FE data for the deep learning network model are usually input in an image form. This is because the values of the calculated results of each pixel are location-dependent and can be stored in the image matrix. Thus, convolutional neural networks (CNN) were typically adopted as the architecture for the training in the FEA surrogating problem. Ronneberger et al. [20] proposed U-Net, which is a convolutional-based network, to segment biomedical images, where the expansive path is symmetric to the contracting part, generating a U-shaped architecture. More importantly, the feature channels in the upsampling paths (skip connection) allow for propagating the context to the higher resolution layers, which leads to a more precise segmentation, and thus, fewer training images are required. This architecture can also be extended for image-generation tasks [21,22]. Esser et al. [23] proposed a variational U-Net for conditionally generated images, where the variational autoencoder preserved the features and provided scalability and robustness in the multiple numerical dimensions.

In the current study, we proposed a DLN framework to predict the history of tissue differentiation for the cases with different dental implants and patient ages, genders and occlusal forces. The DLN framework consists of a conditional variational U-Net-based network and ANN model, where the former substitutes FEA to predict strain, fluid velocity and stem cell concentration, and the latter serves as a substitution of the mechano-regulation algorithm to identify the tissue type of each element in the model. The DLN model improves calculation efficiency while maintaining theoretical prediction accuracy. The training and testing data of the DLN includes the tissue differentiation process calculated by FEA throughout 35 differentiated days with 115 various implant geometries (with 26 geometric parameters; see Appendix H) and 36 types of boundary conditions corresponding to the patients’ ages, genders, and occlusal forces (detailed in Section 4). Note that the healing time period of 35 days was chosen according to the animal tests [24] After the training, the DLN can successfully predict the iterative tissue differentiation process and the growth of bone material properties in real time. The prediction accuracy and efficiency will also be analyzed and discussed.

## 2. Results

### 2.1. Network Overview

The framework was schematically shown in Figure 1, where the deep learning network (DLN) framework is composed of three models. All the models were implemented by using Tensorflow and Python packages. Firstly, Network 1 was built to surrogate the finite element (FE) calculation, which predicts the mechanical response in the model by inputting the information of material properties and occlusal force. Details of the input data format of the occlusal force can be referred to in Appendix A (Figure A1). Network 1 is a conditional variational U-Net that can deal with the various levels of bone properties and occlusal forces. Network 1 extracts the features from the inputs in image format with a size of 128 × 64 by using convolution. The images contain information on five essential properties, including Young’s modulus, Poisson’s ratio, permeability, stem cell concentration, and occlusal force. The outputs are also the images with a size of 128 × 64, containing six mechanical responses including principal strains I, II, III, fluid flow in X and Y directions, and the stem cell concentration for the next iteration. Details of the network architecture can be found in Appendix B (Figure A2).

Secondly, Network 2 (artificial neural network, ANN) aimed to replace the mechano-regulation algorithm, which classifies the differentiated tissue phenotypes based on the mechanical responses obtained from Network 1. Note that the prediction of ANN architecture is pixel-wise, and the dimensions of the input and output in Network 2 are both 1 × 5. The input of Network 2 includes principal strains I, II, and III, and fluid flow in X and Y directions, while the output is the probability of being classified as each tissue phenotype (fibrous tissue, cartilage, immature bone, mature bone, and resorption). Note that the stem cell concentration was not included in the training of Network 2 because it is assumed that the diffusion of the stem cell is independent of the tissue differentiation. On the other hand, the focal loss function [25] was adopted to address the issue of data imbalance because it was found that 90% of the tissue was mature bone. The parameters in the function were determined based on Bayesian optimization [26].

After the training of Networks 1 and 2, tissue differentiation around the implant can be predicted. However, it was observed that some unrealistic results may occur, such as two tissue types forming a checkerboard pattern, or a singular element classified as the cell type that is different from the type of most of its surrounding elements. Thus, we further adopted a weak classifier to reinforce the tissue phenotype classification by using a random forest algorithm [27]. Finally, bone properties changed with the growth of tissue phenotype. The effective material properties for the calculation of the next iteration were then obtained according to stem cell concentration and smoothing procedure, which was based on the iterative process proposed by Chou et al. [10], where the bone properties mapping procedure is detailed in Appendix C. The procedure of tissue differentiation prediction is schematically shown in Figure A3.

### 2.2. The Performance of Mechanical Responses Regression—Network 1

Network 1 aims to replace the FE calculations, which can predict the image of the six mechanical responses for the next iteration based on the input images of four material properties and occlusal force. The performance of Network 1 is evaluated by Pearson’s correlation, where the correlation coefficients of the mechanical responses, i.e., principal strains I, II, III, fluid flow in X and Y directions, and stem cell concentration, were 94.2%, 93.0%, 97.6%, 98.0%, 98.2%, and 99.997%, respectively. In addition, we randomly selected 100 predicted tissue differentiation results throughout 35 days in the 180 testing cases and performed the correlation plot to illustrate the performance of Network 1, as shown in Figure 2, where the colors blue, red, and yellow in the scatter plot represent the data of the pixels located in the tissue differentiation region (callus region), implant, and non-tissue differentiation region (cancellous and cortical bone), respectively. It can be observed that Network 1 can accurately predict the mechanical responses and stem cell concentration. Moreover, even for the cases with some extreme values that were less likely covered in the training data, Network 1 can still give accurate predictions.

Note that the elements of the dental implant were set as not permeable (see Table A1), and thus, their values of fluid flow and stem cell concentration (green dots in Figure 2d–f) were all zero. Similarly, elements in the non-tissue differentiation region had no stem cell diffusion, and thus, the orange dots in Figure 2f accumulated around zero. The results indicate that the prediction conducted by Network 1 has no significant bias, showing a high regression performance.

### 2.3. The Performance of Tissue Phenotypes Classification—Network 2 and Weak Classifier

Network 2 predicted the differentiated tissue phenotype based on the five predicted mechanical responses from Network 1, which aim to surrogate the mechano-regulation algorithm (detailed in Appendix C). The overall accuracy score of testing datasets was 96.24%, as shown in Figure A4b. However, the misclassification rate between mature bone and resorption was still not satisfactory (around 47%). Therefore, a random forest model was applied as a weak classifier to reinforce the tissue phenotypes classification. With the help of the weak classifier, the overall testing classification accuracy reached 97.23%, and the true positive rate of resorption in the confusion matrix was enhanced by 15%, as shown in Figure A4c.

### 2.4. Day-By-Day Continuous Tissue Differentiation History Prediction

With the outputs of Networks 1 and 2, day-by-day continuous results of tissue differentiation around the implants can be predicted. Here, the histogram analysis of the accuracy of the tissue phenotype distribution on the 35th day after the day-by-day continuous prediction for all the implants in the dataset is shown in Figure 3. It was found that the distribution of training and testing were similar. Most of the accuracy scores in both cases were in the range of 75 to 93%, showing the robustness of the current developed framework.

We next visualized the day-by-day continuous predicted results of two randomly selected implants (referred to as Implants A and B) with different geometries, bone material properties, and occlusal forces, as shown in Figure 4. The healing pattern in both the former (the 5th day) and later stages (the 35th day) was presented to show the proposed DLN’s efficacy. Three features that were generated by finite element analysis (FEA) (i.e., ground truth) were captured and predicted by the current DLN in both implants. First, immature bone formed a band-like structure that connected the thread tips in the former stages. Second, the cartilage occurred in the bottom of the implants. Third, mature bone dominated the callus region while some resorption occurred around the threads. The misclassification pixels between the prediction and the ground truth were labeled in red while the white pixels representing the pixels were accurately classified by the current DLN. It can be observed that, although some pixels around the threads and the bottom were misclassified, most of the pixels were accurately predicted. The overall prediction accuracy was 82%.

Next, receiver operating characteristic (ROC) curves and the corresponding area under the curve (AUC) values of day-by-day continuous prediction results were determined to further evaluate the prediction performance of the current DLN, as shown in Figure 5. It was found that, in the testing cases, the AUC values of the five tissue phenotypes, i.e., fibrous tissue, cartilage, immature bone, mature bone, and resorption, were 0.99, 0.97, 0.89, 0.88, and 0.86, indicating a good overall classification accuracy.

From the perspective of image analysis in clinical treatment and animal experiments, two important performance indexes are typically used for analyzing the osseointegration between bone and the implant. They are bone area (BA) and bone–implant contact (BIC), where BA is defined as the percentage of the total area of the mature/immature bone elements in the area between all the threads; BIC is defined as the percentage of the total length of the interface between the implant elements and the mature/immature bone elements to the total length of all the threads. Note that the two indexes were obtained from the tissue phenotype distribution on the 35th day predicted by the current DLN, rather than obtained from the direct output of the DLN. Figure 6 shows the correlation of performance indexes between the predicted and the ground truth of 10 randomly selected testing cases. The testing correlation coefficients of BIC and BA values were both 85%. This indicates that the current DLN can provide implant design guidelines similar to those recommended by the mechano-regulation algorithm.

## 3. Discussion

To sum up, a DLN model that consists of two networks and a weak classifier (random forest) was proposed to predict the tissue differentiation throughout 35 days around different dental implants in the patient with different ages, genders, and occlusal forces. The first network based on the conditional variational U-Net predicted the six essential mechanical responses. The testing correlation coefficients of the six essential mechanical responses were all higher than 93%. The training and testing datasets both include various implant geometries, bone properties, and occlusal force (FE data generation is detailed in Section 4.1). This implied that the changes in these parameters could be accurately captured by Network 1. Network 2 classifies the tissue phenotypes with an overall testing accuracy of around 96.24%. It is remarkable that the accuracy is very good, even the input, i.e., mechanical responses, with the prediction error from Network 1. The current framework is proven to be effective. Finally, the mechanical responses predicted by Network 1 and the probability of tissue phenotypes by Network 2 were inputted into the random forest model. The accuracy then reached 97.23% as more information was provided for the classification.

For the results of day-by-day continuous tissue differentiation prediction, it was found that the testing AUC values of each tissue phenotype throughout the 35 days were all over 0.86. The correlation coefficient of the two performance indexes, i.e., BA and BIC, between the predicted results and the FEA results were both around 85%. This implies that the current DLN model can reproduce the FEA results very well. However, it can be found that the mature bone and resorption were likely to be misclassified compared to other cell phenotypes. This is because the values of biophysical stimulus criteria of the two tissue phenotypes were very close (see Table A1). For these reasons, the predicted performance indexes (BA and BIC) tended to be overestimated, as shown in Figure 6. In general, the DLN model still shows its robust performance in predicting the tissue differentiation process. The prediction performance can always be further improved by increasing the datasets.

The calculation times of FE data generation, DLN training, and DLN inference over all 900 cases were recorded and are shown in Figure 7. Typically, it takes an hour to predict the tissue differentiation results throughout 35 days by using FEA, while the DLN model only needs about 50 **s** per case. Even if the DLN training time is taken into account, the total required time for the DLN training and inference is still significantly lower than the time needed for the FEA calculation. Therefore, it is proven that the DLN model can significantly accelerate the prediction process while maintaining prediction accuracy.

Moreover, the main contribution of the proposed DLN model is its great scalability and generality. This allows the DLN model to infer the tissue differentiation process based on any values of material properties and occlusal forces that fall in a similar range to those included in the dataset. Note that the range of the ratio of the bone properties and the magnitude of the occlusal force in the dataset was from 0.42 to 1.00 and from 75 to 139 N (see Table 1 for more details).

Figure 8 shows two cases where the selected implant geometries, bone properties, and the magnitudes of the occlusal force were not included in the training and testing dataset. The ratio of bone properties in Cases 1 and 2 was set to be 0.5 and 0.95 while the occlusal force was set to 90 and 120 N, respectively. The overall accuracy throughout 35 days of Case 1 and Case 2 was 79% and 84%, respectively. Although the accuracy of the prediction dropped slightly compared to the accuracy in the training and test datasets, the trend of the tissue differentiation procedure and the bone healing features of the two cases can still be captured, for example, the immature bone band connected between threads in the early stage of Case 1, the immature bone occupied almost all regions between the threads in the early stage of Case 2, the mature bone dominated the tissue differentiation region on Day 35, etc. The results of the two case studies proved that the proposed DLN model can be extended to cases with various implant geometries, bone properties, and magnitudes of the occlusal force, corresponding to the patient’s personal conditions.

The proposed DLN framework is based on the mechano-regulation algorithm, and it can well predict bone healing based on personal conditions and can assist in the process of engineering design for dental implants. Its applicability is greatly improved compared with the methods used in other works. For example, the prediction efficiency is significantly improved compared to the work conducted by Li et al. [1], so that the prediction of bone healing can be achieved in real time. In contrast, Marin et al. [24] performed a conventional in vivo animal test to examine the effect of bone ingrowth in various implant designs, which took 35 days and required extensive preparation to obtain histomorphologic images. The proposed DLN model is extended to be personally predicted under a variety of conditions while the DLN model proposed by Hsu et al. [22] focuses on single boundary conditions and attributes to the relationship between geometry design and bone healing.

However, the current model is established under the following assumptions and limitations. First, in this study, we assumed that the mechanism of bone healing can be well described by the current mechano-regulation theory. There were some more factors that affect bone healing that can be considered, for example, random-walk stem cell migration [28], extending the model from two-dimensional axis-symmetry to three-dimensional [29], toxicological concerns [30], chemical compounds application such as nano-antioxidants [31] for enhancing bioactivity, and health concerns such as infection [32] and allergic reaction [33]. Second, the prediction accuracy can be further improved by optimizing all the hyperparameters and network architecture. Third, the prediction performance of personal conditions (i.e., bone strength, occlusal force, etc.) that are out of the range of the training dataset remains unexamined at this stage. Fourth, to apply the proposed framework to clinical use, accurate bone strength and occlusal strength of patients need to be measured for better prediction results. These limitations are considered future work of the current work, since considering too many extensions may be out of the scope of this work.

Note that the health concerns around implants are crucial for bone healing and vary within individuals. The proposed DLN framework enables efficient prediction of tissue differentiation based on the input mechanical properties depending on patient’s age, gender and conditions. Thus, it is possible for the current DLN to take the health concerns of patients into account, provided that the health status can be reflected by the input mechanical properties. For example, infection or allergic reactions may cause softening of local tissues, which may decrease the input value of the Young’s modulus. However, as the mechano-regulation algorithm does not consider any health concerns mentioned above, the model required further improvement to achieve a more realistic prediction.

The proposed framework has great potential to be applied for clinical use. It allows for generating an implant according to personal conditions on the fly and can predict the personal bone healing process before insertion. This could break the traditional limitation of fix-sized implants as well as improve the healing performance. In this study, we performed two case studies as a proof of concept, showing the robustness of the proposed framework. Our proposed framework has the potential to significantly reduce the cost and time of the design procedure of implants, which typically requires mechanical and animal tests.

Finally, the current study did not focus on revealing the effects between geometry design and the performance indexes, i.e., to explain and attribute the reasons to why the neural network outputted the prediction and classification. Attribution techniques, such as deep Taylor decomposition (DTD) [34] and layer-wise relevance propagation (LRP), are commonly used for interpreting neural networks. They can explain the prediction results by redistributing the relevance between input and output data. Among them, the DTD technique is particularly suitable to be applied to the current work to attribute the relationship between implant geometry and performance indexes and to provide features/design guidelines for dental implants that are difficult to observe. This can push the application of the current DLN to the next level. In addition, the main contribution of the current work is to develop a DLN that enables over 90× reduction in computation time for predicting tissue differentiation and bone healing affected by implant geometry, patient age, gender, and occlusal forces. Thus, the DLN can not only optimize the implant geometry but can also serve as an efficient tissue engineering tool to manipulate tissue differentiation and healing patterns by virtual trial and error with low cost.

## 4. Materials and Methods

### 4.1. Dataset Preparation by FEA

Training and testing dataset of tissue differentiation was generated based on mechano-regulation theory. All the calculations and iterative procedures were implemented by using ANSYS (Canonsburg, PA, USA) and MATLAB (MathWorks, Natick, MA, USA). The differentiated tissue phenotypes are determined by the biophysical stimulus, which is calculated from principal strains I, II, III, and fluid flow X, Y (see Equation (A1) in Appendix C). Then, bone properties changed correspondingly as the cells matured, i.e., stem cell concentration became higher. By repeating the procedure, tissue differentiation history and bone property change can be obtained. Details of the iterative process and rules of tissue differentiation can be found in Appendix C. In addition, the FE model was validated by comparing the animal test results by Marin et al. [24] and demonstrated in the author’s previous work [1], as shown in Figure A5.

Figure 9 shows the model and boundary condition adopted in the current study, as well as the dimension and layout of each type of material in the bone–implant system. Note that the material properties of bones and each tissue phenotype can be referred to in Table A2. Most implant geometries can be described by 26 parameters, such as platform-switched, double-threaded, and tapered body design. The definition and schematic of the 26 parameters (*P1–P26*) are shown in Appendix H (Figure A5 and Table A3). The radius of the implant head and body are controlled by *P1* and *P2*. For example, a platform above the first thread was formed when *P1* < *P2*. Based on the literature [1], the design of the platform is beneficial to minimize the maximum of von Misses, compressive, and tensile stresses because it can share the occlusion force applied from the top of the implant. *P3* is the distance from the top to the first upper thread; *P5* is the distance between two upper threads; *P7-P11* describe the geometry of the upper threads, as shown in Figure A5b; *P25* is the number of the upper threads. Next, considering the parameters related to the lower threads, *P4* is the distance from the top to the first lower thread; *P6* is the distance between two lower threads; *P12–16* describe the geometry of the lower threads, as shown in Figure A5c, and *P26* is the number of the lower threads. Finally, *P17-P24* represent a curve of a cubic function, which alters the implant body shape, e.g., an implant with a tapered structure. Note that based on the literature [15], both the double-threaded and tapered body designs are beneficial for easier insertion of the implants. In the current study, 115 implants were generated by randomly assigning the 26 parameters for the DLN training and testing.

Figure 9a shows the boundary condition of the bone–implant system, where the occlusal force was applied on the top surface of the implant; the bottom of cancellous bone was set to be fixed. Note that the current work aims to improve the prediction efficiency and generality of the prediction by the DLN framework. Thus, training and testing datasets with various bone properties and occlusal forces were included, representing different health conditions of the patients, i.e., age and gender, bone properties and occlusal force. Here, in order to reduce the complexity of DLN training, six categories based on age and gender of the patients were defined instead of considering continuous changes in bone property values. They are 15-year-old young women, 30-year-old adult women, 75-year-old older women, 15-year-old younger men, 30-year-old adult men, and 75-year-old older men.

The values of the bone properties for all the categories can be obtained by applying the bone properties ratios. According to the relationship proposed by Carter and Hayes [35], the Young’s modulus of bone is proportional to the cube of the bone density. Thus, the cube of the ratio between average bone density for all six categories can be regarded as the ratios of bone properties [23], as shown in Table 1. Note that here the ratio for adult men was set as 1, and the value of bone properties can be referred to in Table A2. Now, considering the occlusal force, Miyaura et al. [36] reported the average total occlusal force of all the teeth for both gender and different ages. However, our study focuses on the dental implant for single tooth only, and thus, a scaling process is still needed. The total occlusal force of all the teeth of an adult man reported in the literature is 395 N, and the magnitude for a tooth of an adult man is about 139 N [25]. Thus, the magnitude of occlusal force for a tooth for the remaining five categories can be obtained by applying the same scaling factor, as shown in Table 1. Thus, by combining six categories of bone properties and six levels of occlusal force, thirty-six types of boundary conditions can be used to generate the training and testing datasets.

Tissue differentiation was regarded as the short-term bone healing process, where the healing period was defined as the first 35 days after inserting the implant in the current study. Training and testing FE data included the mechanical responses, tissue differentiation history, and the changes in material properties throughout 35 days based on the mechano-regulation algorithm. We randomly selected 900 cases from combinations of 115 implant geometries and 36 types of boundary conditions, serving as the training and testing datasets. The training and testing domain were cropped into 3.2 × 6.4 mm, as shown in Figure 9b. Then, the images were further interpolated into a pixel-wised array with a size of 128 × 64 by a linear function.

### 4.2. Neural Network Settings and Training

The hyperparameters in Networks 1, 2, and the random forest model were optimized by Bayesian optimization. The optimized hyperparameters in Network 1 were the number of filters, filter size, and activation function; those in Network 2 were the node number of each dense layer, the number of dense layers, and the parameters in the focal loss function; those in the random forest model were the estimator (tree numbers) and the max depth (tree depth). The optimizer for both Networks 1 and 2 adopted the Adam optimization algorithm. The loss function of Network 1 consisted of two parts: reconstruction mean absolute error (MAE) and the Kullback–Leibler (KL) divergence:(1)$$L\left(x,\theta ,\phi \right)=-KL((q_{\emptyset }\left.\left(z|\hat{y}\right)\right\|p_{\theta }\left(z|x,\hat{y}\right))+E_{q_{\emptyset }}(\log p_{\theta }\left(x|z,\hat{y}\right)),$$
where $p_{\theta }$ and $q_{\emptyset }$ are probability distributions, $E_{q_{\emptyset }}$ is an expectation value, x is the material properties images input, $\hat{y}$ is the boundary condition input, and z is the essential mechanical responses images output.

The dataset that consists of 900 cases was randomly split into 80% for training (720 cases) and 20% for testing (180 cases). The tissue differentiation history throughout 35 days for each selected case was included in the training dataset. As shown in Figure 1, the input of Network 1 was material properties and the occlusal force; the output of Network 2 was the tissue differentiation results in the next day. The material properties for the next day can be further obtained by the mapping procedure.

## 5. Conclusions

In this study, the proposed deep learning network (DLN) framework can provide a rapid engineering design guideline of dental implants by predicting the history of tissue differentiation throughout 35 days around the implant under various bone properties and occlusal forces (i.e., for patients with different ages and genders). The network successfully surrogated the finite element (FE) calculation and mechano-regulation algorithm and significantly increased the calculation efficiency based on the results shown in Figure 2 and Figure 7. Thus, the performance of implants can be evaluated in less than a minute. The key contribution of the current work is the design of the training dataset and the proposed DLN framework, where the conditional variational U-Net allowed for inference and was scaled to various boundary conditions. The current work is a proof of concept that the DLN model can be an alternative to the complex, time-dependent calculation of finite element analysis, which has the potential to assist in the customization of dental implant treatment and the evaluation the implant performance with high efficiency. The results of the present work are expected to benefit dental treatment and improve the healing performance of patients.

## Figures and Tables

**Figure 1 ijms-24-01948-f001:**
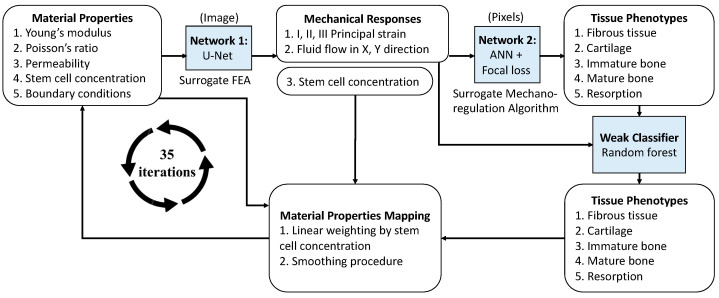
The framework of the deep-learning-based network of short-term bone healing prediction.

**Figure 2 ijms-24-01948-f002:**
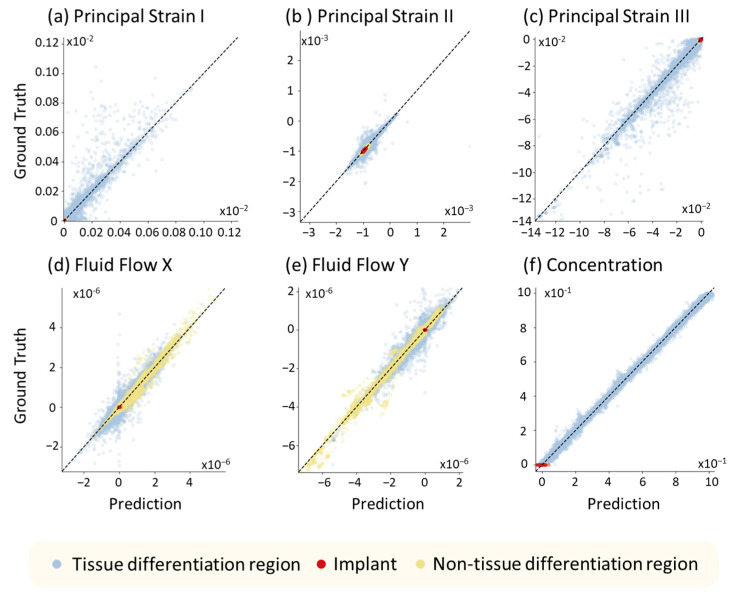
Correlation plot between the predicted values and ground truths calculated by finite element analysis (FEA), where (**a**–**f**) are principle strain I, II, III, fluid velocity in the x and y directions, and stem cell concentration. The data include the elements in the tissue differentiation (blue), implant (red), and non-tissue differentiation (yellow) regions.

**Figure 3 ijms-24-01948-f003:**
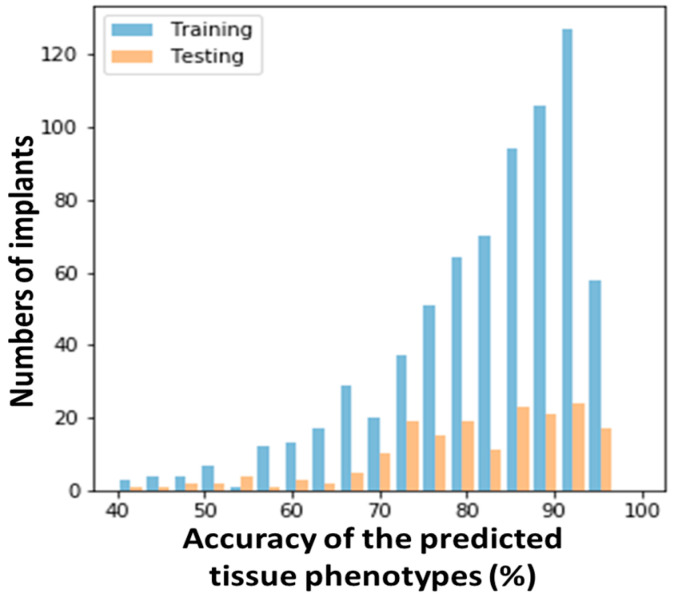
Histogram analysis of the accuracy on the 35th day after the day-by-day continuous prediction for all the implants in the dataset.

**Figure 4 ijms-24-01948-f004:**
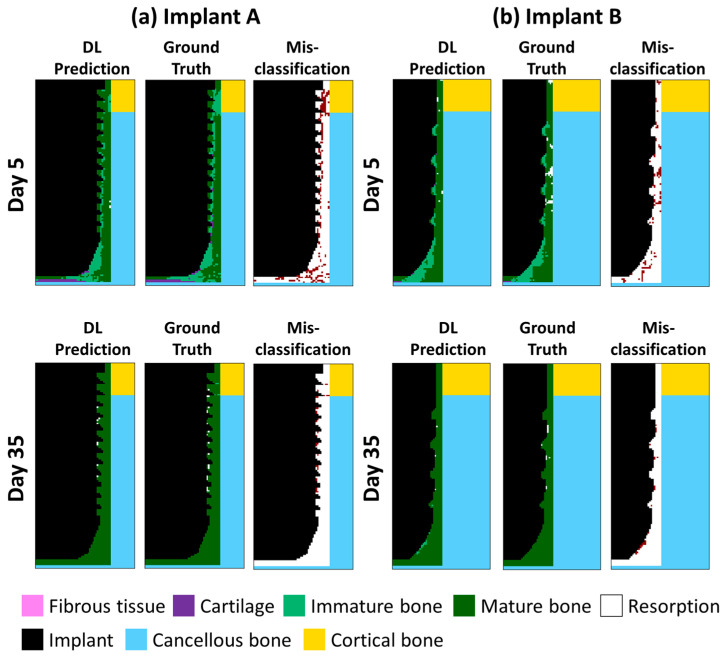
The results of the DLN prediction, ground truth, and the misclassification pixels of two selected implants on days 5 and 35.

**Figure 5 ijms-24-01948-f005:**
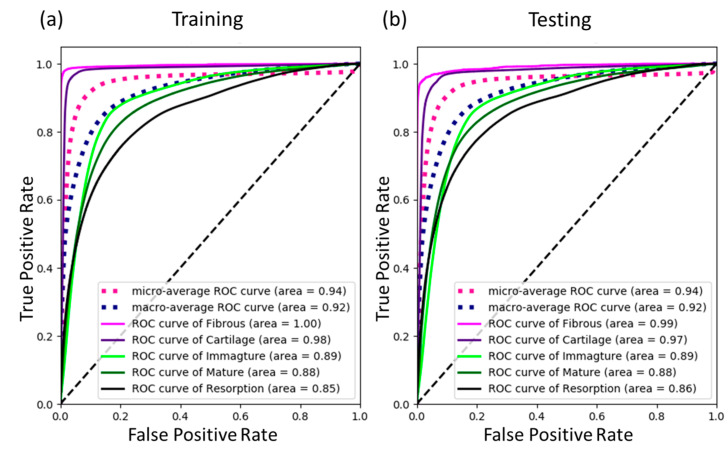
ROC curves and the corresponding AUC values for the day-by-day continuous prediction for each tissue phenotype.

**Figure 6 ijms-24-01948-f006:**
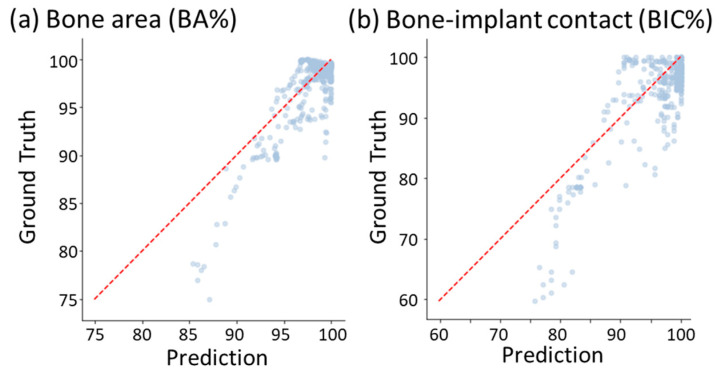
The correlation plot between prediction and the ground truth (FE results) of performance indexes.

**Figure 7 ijms-24-01948-f007:**
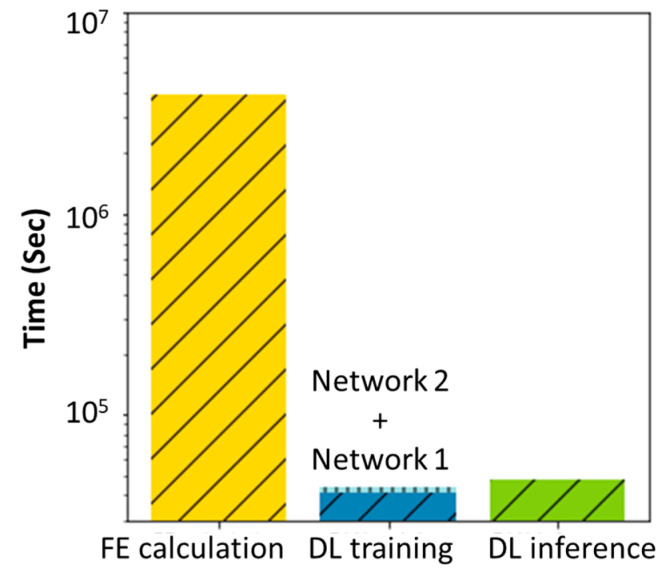
The computational time of tissue differentiation prediction for all 900 cases needed by FE calculation and DLN.

**Figure 8 ijms-24-01948-f008:**
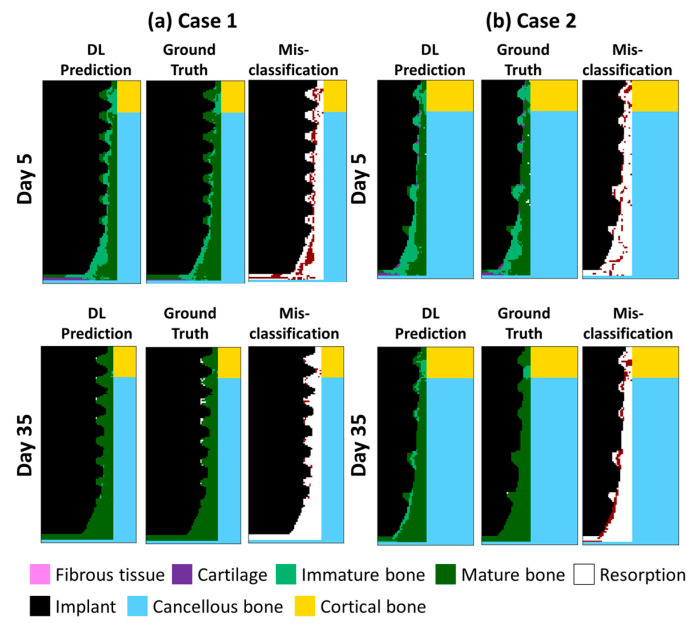
Tissue differentiation results on Day 5 and 35 for (**a**) Case 1 and (**b**) Case 2 with various boundary conditions.

**Figure 9 ijms-24-01948-f009:**
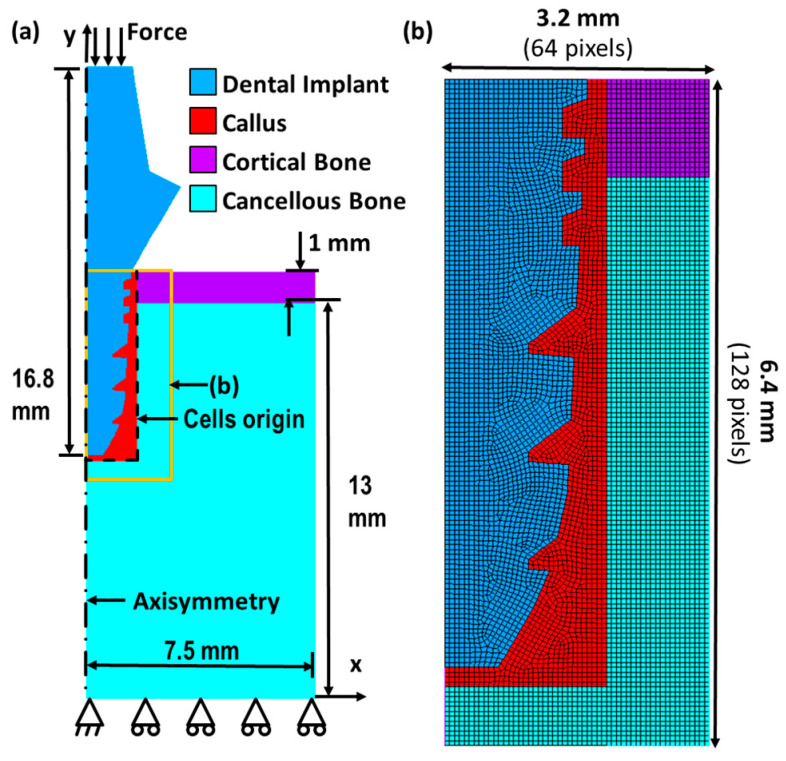
The schematic diagrams of (**a**) the FE model and (**b**) the cropped domain for the DLN training and testing.

**Table 1 ijms-24-01948-t001:** The bone properties and occlusal forces used in the current study.

	Young Women	Adult Women	Older Women	Young Men	Adult Men	Older Men
The ratio of bone properties	0.78	0.88	0.42	0.86	1	0.75
Occlusal force (N)	100	105	62.5	114	139	75

## Data Availability

Not applicable.

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
