# Peer review of "Prediction of Bone Healing around Dental Implants in Various Boundary Conditions by Deep Learning Network"

_ijms, 2023, doi:10.3390/ijms24031948_

Round 1

Reviewer 1 Report

1. The main question addressed by the research is the in deep learning and silico models for bone healing around the dental implant and this work is promising for future tissue enginering of Dentistry. 

2. The topic is original in the field and it will better to compare with other invitro and clinical studies. 

3. Please address a specific gaps in the health concerns of implants in discussion section. 

4. Please add what does this work add to the tissue engineering area compared with other published material?

5.The authors should consider the limitations of deep learning regarding the methodology.

6.the authors should add new published works in tissue enginering using in silico and artificial intelligence and also in vitro works in dental implant studies. 

7.Conclusion need to be consistent with the evidence and arguments presented. 

8.Please include additional schematic figures to simplify the understanding of main points of work.

Reviewer 2 Report

dear authors 

congratulations on your study. A few concerns I have -

limitations are not mentioned

methods are written in the end it should be shifted before results.

list all abbreviations used in the end 

change references to the new ones very old references you have used.

modify the citation to the Vancouver system- recheck journal requirements. you have cited irregularly.

concise the appendix - too many and too much details. explain the in your manuscript that will be more adequate. 

please add the clinical significance of the study
